# Molecular Interaction Studies and Phytochemical Characterization of *Mentha pulegium* L. Constituents with Multiple Biological Utilities as Antioxidant, Antimicrobial, Anticancer and Anti-Hemolytic Agents

**DOI:** 10.3390/molecules27154824

**Published:** 2022-07-28

**Authors:** Aisha M. H. Al-Rajhi, Husam Qanash, Mohammed S. Almuhayawi, Soad K. Al Jaouni, Marwah M. Bakri, Magdah Ganash, Hanaa M. Salama, Samy Selim, Tarek M. Abdelghany

**Affiliations:** 1Department of Biology, College of Science, Princess Nourah bint Abdulrahman University, P.O. Box 84428, Riyadh 11671, Saudi Arabia; amoalrajhi@pnu.edu.sa; 2Department of Medical Laboratory Science, College of Applied Medical Sciences, University of Ha’il, Ha’il 55476, Saudi Arabia; h.qanash@uoh.edu.sa; 3Molecular Diagnostics and Personalized Therapeutics Unit, University of Ha’il, Ha’il 55476, Saudi Arabia; 4Department of Medical Microbiology and Parasitology, Faculty of Medicine, King Abdulaziz University, Jeddah 21589, Saudi Arabia; 5Department of Hematology/Oncology, Yousef Abdulatif Jameel Scientific Chair of Prophetic Medicine Application, Faculty of Medicine, King Abdulaziz University, Jeddah 21589, Saudi Arabia; saljaouni@kau.edu.sa; 6Biology Department, Faculty of Science, Jazan University, Jazan 82817, Saudi Arabia; marwah890@gmail.com; 7Biology Department, Faculty of Science, King Abdulaziz University, Jeddah 21589, Saudi Arabia; mganash@hotmail.com; 8Department of Chemistry, Faculty of Science, Port Said University, Port Said 42521, Egypt; hana_negm20010@yahoo.com; 9Department of Clinical Laboratory Sciences, College of Applied Medical Sciences, Jouf University, Sakaka 72341, Saudi Arabia; 10Botany and Microbiology Department, Faculty of Science, Al-Azhar University, Cairo 11884, Egypt

**Keywords:** *Mentha pulegium*, anticancer, anticoagulant, antimicrobial, antioxidant

## Abstract

Multiple biological functions of *Mentha pulegium* extract were evaluated in the current work. Phytochemical components of the *M. pulegium* extract were detected by Gas Chromatography-Mass Spectrometry (GC-MS) and High-performance liquid chromatography (HPLC). Moreover, *M. pulegium* extract was estimated for antioxidant potential by 2,2-Diphenyl-1-picryl-hydrazyl-hydrate (DPPH) free radical scavenging, antimicrobial activity by well diffusion, and anticoagulant activity via prothrombin time (PT) and activated partial thromboplastin time (APTT). GC-MS analysis detected compounds including cholesterol margarate, stigmast-5-en-3-ol, 19-nor-4-androstenediol, androstan-17-one, pulegone-1,2-epoxide, isochiapin B, dotriacontane, hexadecanoic acid and neophytadiene. Chrysoeriol (15.36 µg/mL) was followed by kaempferol (11.14 µg/mL) and 7-OH flavone (10.14 µg/mL), catechin (4.11 µg/mL), hisperdin (3.05 µg/mL), and luteolin (2.36 µg/mL) were detected by HPLC as flavonoids, in addition to ferulic (13.19 µg/mL), cinnamic (12.69 µg/mL), caffeic (11.45 µg/mL), pyrogallol (9.36 µg/mL), *p*-coumaric (5.06 µg/mL) and salicylic (4.17 µg/mL) as phenolics. Antioxidant activity was detected with IC_50_ 18 µg/mL, hemolysis inhibition was recorded as 79.8% at 1000 μg/mL, and PT and APTT were at 21.5 s and 49.5 s, respectively, at 50 μg/mL of *M. pulegium* extract. The acute toxicity of *M. pulegium* extract was recorded against PC3 (IC_50_ 97.99 µg/mL) and MCF7 (IC_50_ 80.21 µg/mL). Antimicrobial activity of *M. pulegium* extract was documented against *Bacillus subtilis*, *Escherichia coli*, *Pseudomonas*
*aureus*, *Candida albicans*, *Pseudomonas aeruginosa*, but not against black fungus *Mucor circinelloides*. Molecular docking was applied using MOE (Molecular Operating Environment) to explain the biological activity of neophytadiene, luteolin, chrysoeriol and kaempferol. These compounds could be suitable for the development of novel pharmacological agents for treatment of cancer and bacterial infections.

## 1. Introduction

At the present time, with the spread and outbreak of diseases, and the resistance of pathogens to drugs, scientists in the medical and pharmaceutical fields have increased their search for safe alternative sources that can solve these problems [1]. Plants are one of the best sources of herbal preparations because they contain different and diverse phyto-constituents. Despite many biologically active compounds of plant origin having been discovered, there are many that so far remain undiscovered. There are many medicinal plants that have been traditionally used without any scientific basis, requiring further study for the conversion from traditional use to scientific validity [2,3].

*Mentha* (commonly named pennyroyal) of the family *Lamiaceae* contains 61 species with approximately 100 varieties which are disseminated across temperate regions including North America, Africa, Asia, Europe, and Australia [4]. The family *Lamiaceae* is considered a rich source of flavonoids, particularly flavones such as apigenin, luteolin and 6-hydroxyflavone [4]. According to Marzouk et al. [5] the genus *Mentha* is represented in the flora of Egypt by two species, namely *Mentha longifolia* and *M. pulegium* L. Numerous uses have been recorded for the *M. pulegium* plant [6] including for relief of colds, coughs, headaches, kidney problems, liver complaints, and gallbladder ailments; in addition, it has been applied in drink and food additives. Despite these therapeutic and nutritional uses, *M. pulegium* is well-known for its poisonousness [4]. Additionally, some reports have indicated the presence of certain toxic compounds in *M. pulegium*, although Caputo et al. [7] suggested that the soil conditions may influence the expression of toxic compounds such as pulegone and its derivatives. Numerous biological activities have previously been reported for *M. pulegium* extract as well as its essential oils, and occasionally the activity has differed based on cultivation conditions. Previous studies reported a number of biological activities including antimicrobial activity against many microorganisms [8,9], decremental inflammation with oral mucosa treatment, anti-hemolytic, and anticancer activities [10] of Mentha.

Recently, Alharbi et al. [11] demonstrated the bactericidal potential of *M. pulegium* extract towards four gram-positive and five gram-negative bacteria; also, the development of *Aspergillus niger*, *A. flavus* and *Candida albicans* was inhibited by *M. pulegium* extract. Baali et al. [9] suggested food and pharmaceutical applications of Algerian *M. pulegium* as a preservative, antimicrobial, and antioxidant agent. Moreover, the anthelmintic properties of *M. pulegium* extract were recently documented [12]. Some scientific papers have focused on the application of *M. pulegium* essential oils [13,14] in biological activities including antiviral and antifungal activities. Bektašević et al. [15] observed excellent antioxidant activity in addition to moderate cholinesterase inhibition using essential oils, and therefore recommended their use in Alzheimer disease. The most serious diseases worldwide in the last decade have been cancers, which in 2020 led to about 9.9 million deaths [16]. Promising sources of phenolic and flavonoid contents in plants can play a main role in cancer treatment and treatment of other diseases [17,18]. The inhibitory effect of plant flavonoids on the spread of cancer cells through apoptosis induction has been detected [19,20]. The existence in *M. pulegium* of several phenolic acids was reported, including syringic acid and ferulic acid, as well as several compounds of flavonoid including isorhamnetin-3-*O*-glucoside and kaempferol-3-*O*-rutinoside; these compounds reflected high antioxidant activity [12]. Positive correlations among the phenolic contents of certain *Mentha* species (*M. rotundifolia* and *M. pulegium*) and their biological activities, namely antimicrobial and antioxidant activities, were documented by Alharbi et al. [11].

A computer simulation technique such as molecular dynamic simulation (MDS) can be utilized [20] to screen and estimate the physical movements of molecules and atoms. This allows searchers to measure flexibility, rigidity, and secondary construction prediction in terms of gain or loss during the simulation time. Recently, the silico protocol was applied to select the most suitable structures for biological observation of active compounds; this protocol was dependent on the correlation among biological activity values of the input structures and the target proteins in pathogenic organisms, cancer cells, etc. [20,21,22,23,24,25]. The present investigation was designed to explore the phytochemical characterization of *M. pulegium* extract by GC-MS and HPLC, studying different biological applications comprising antitumor, antimicrobial, anti-hemolytic and anticoagulant activities.

## 2. Results and Discussion

### 2.1. GC-MS and HPLC Analysis of M. pulegium Extract

Research into the pharmaceutical value of medicinal plants depends on the detection and identification of phyto-constituents and their chemical structures, and understanding their pharmacodynamic action when used for the treatment of any illness. In the current study, extract of *M. pulegium* was chemically evaluated using GC-MS and HPLC analysis (Figure 1) for performing various biological activities. The GC-MS analysis of *M. pulegium* (Figure 2) revealed the existence of 38 compounds (Table 1) and, as indicated by chromatogram (Figure 2), the most abundant component by peak area was shown to be hexadecanoic acid (20.88%), followed by methyl 10-ketostearate (10.84%), cis-13-Octadecenoic acid (6.84%), stearic acid (4.87%), oxiraneoctanoic acid 3-octyl-,methyl ester (4.72%), methyl hexadecanoate (4.56%), elaidic acid, methyl ester (4.17%), and 1,2-benzenedicarboxylic acid (3.60%). Neophytadiene (diterpenoid) was identified as a constituent of *M. pulegium* extract (Table 1); this constituent has been found to include multiple functionalized agents according to to earlier reports [26,27], including anti-inflammatory, analgesic, antipyretic, antioxidant, and antimicrobial. Among the *M. pulegium* extract constituents, phytosterols including cholesterol margarate, stigmast-5-en-3-ol, 19-nor-4-androstenediol, and androstan-17-one (Table 1), can play an important biological role in protection against various cancers of the stomach, breast, prostate, lung, liver, and ovaries, as well as regulation of sugar uptake in diabetics, as mentioned by previous researchers [28,29,30]. Raju et al. [31] documented the safe properties (non-carcinogenic, non-poisonous, and non-mutagenic) of these phytosterols, and the anti-hypercholesterolemic qualities of stigmast-5-en-3-ol have also been reported [32]. Furthermore, the obtained results showed the presence of other compounds in *M. pulegium* extract, namely pulegone-1,2-epoxide (oxygenated terpene), which has vasorelaxant activity according to Lima et al. [33]. The non-hemolytic, non-toxic, and non-irritant qualities of pulegone-1,2-epoxide have been recognized [34], therefore its application for medicinal purposes was validated. Other important ingredients of *M. pulegium* extract (Table 1), namely isochiapin B (sesquiterpene lactone) and dotriacontane, have been identified as antioxidants [35] and bactriocidal agents [36], respectively. From the GC-MS, it is obvious that *M. pulegium* extract is rich in biologically active constituents. Different and various flavonoids and phenolic acids within *M. pulegium* extract were recognized by HPLC analysis (Table 2 and Figure 3 and Figure 4). Among the detected flavonoids, chrysoeriol was the most abundant(15.36 µg/mL), followed by kaempferol (11.14 µg/mL), and 7-OH flavone (10.14 µg/mL) (Table 2), while luteolin, hisperdin, and catechin were detected in the extract with low concentrations of 2.36, 3.05, and 4.11 µg/mL, respectively. From a scientific study [37] of thirteen *Mentha* species, the detected flavonoid levels varied within the species, particularly in the incidence of hesperidin, luteolin, and kaempferol, which ranged from 0.73 to 109.39 µg/g, 1.84 to 31.03 µg/g, and 1.30 to 33.68 µg/g, respectively; in contrast, levels of phenolic acids did not significantly differ. Six phenolic acids were identified in the *M. pulegium* extract, namely ferulic, cinnamic, caffeic, pyrogallol, *p*-coumaric, and salicylic with concentrations of 13.19, 12.69, 11.45, 9.36 5.06 and 4.17 µg/mL, respectively. Similar observations were recorded in a previous study [38], where caffeic acid represented the main compound (31.48 mg/g) in *M. pulegium*. Different flavonoids and phenolic acids were detected in *M. pulegium*, depending on the origin, as mentioned previously in the literature. Caffeic (as phenolic acid), acacetin 5-*O*-α-l-rhamnopyranosyl(1-2)-*O*-α-l-rhamnopyranoside, 7-*O*-α-rutinosides of apigenin and luteolin, vicenin, and 5-hydroxy-6,7,3′,4′-tetramethoxyflavone as flavonoids were detected in *M. pulegium* of Egyptian origin [39]. Caffeic, vanillic, and ferulic as phenolic acids, in addition to apigenin, luteolin, naringenin, and catechin as flavonoids, were detected in *M. pulegium* of Greek origin [40]. 4-Hydroxy benzoic, caffeic, *p*-coumaric, chlorogenic, and rosmarinic acids as phenolic acid, in addition to luteolin, diosmin, and kaempferol, were detected in *M. pulegium* of Algerian origin [41]. Hesperidin was recorded in *M. pulegium* at low concentrations compared with the other detected flavonoids (Table 2), was although it has been recorded in various levels in other species, for example 0.21 mg/g in *M. piperita* and 11.83 mg/g in *M. spicata* [42]. The main reason for the traditional therapeutic potential of *Mentha* spp. may be due to the existence of these flavonoids and phenolics that provide different health benefits. In a recent evaluation [43], multiple biological functions including anticancer, antioxidant, antimicrobial and anti-inflammatory properties of chrysoeriol and luteolin were documented. Moreover, Wei et al. [44] demonstrated in vivo activity of chrysoeriol against human lung carcinoma development. Cancer-preventive action without distracting normal cells and anti-inflammatory activities of kaempferol were recorded by Sharma et al. [45], in vivo and in vitro. Recently, Abbou et al. [46] identified different compounds, namely *p*-coumaric acid, gallic acid, naringenin, quercetin, and ferulic acid, in the extract of *M. pulegium* L aerial parts, which could be supportive in the regulation of oxidative stress leading to Alzheimer’s disease and diabetes.

### 2.2. Antimicrobial Potential of M. pulegium Extract

Antimicrobial activities of *M. pulegium* extract were assessed against four bacteria, one unicellular fungus, and one black mold as presented (Table 3 and Figure 5). The findings indicated that all the tested bacteria and *C. albicans* were sensitive to *M. pulegium* extract. Growth of *B. subtilis* (IZ, 27 mm) was more affected by *M. pulegium* extract than other bacteria, namely *E. coli* (IZ, 26 mm), *S. aureus* (IZ, 19 mm) and *P. aeruginosa* (IZ, 19 mm). Surprisingly, the extract was more effective on the tested organisms than the positive control (Gentamycin antibiotic). However, *C. albicans* was inhibited where the IZ was 25 mm, while *Mucor circinelloides* growth was not inhibited. These results may be due to differences in cell wall composition or genes responsible for extract resistance. According to a recent study [47], *M. pulegium* extract exhibited antibacterial potential against *S. aureus* only, and therefore it was applied for wound healing. Previous reports have indicated the antimicrobial activity of *M. pulegium* extract towards many bacteria (including five gram-positive and five gram-negative types) and fungi including six fungal species [7]. Numerous studies have shown that *M. pulegium* essential oil exerts antimicrobial activities on bacteria and fungi. A study by Aimad et al. [48] indicated that *M. pulegium* oil completely inhibited *Aspergillus niger* growth, and inhibited *B. subtilis* where the IZ was 25 mm. The antibacterial activity is perhaps attributable to the occurrence of natural ingredients of *M. pulegium* extract including neophytadiene, isochiapin B, and other constituents mentioned in Table 3. The current finding is in good agreement with the results of Ceyhan-Güvensen and Keskin [49], who reported the antimicrobial activity of *M. pulegium* extract and explained this activity by the presence of high neophytadiene content.

TEM of untreated and treated *S. aureus* and *P. aeruginosa* with *M. pulegium* extract are shown in Figure 6. The untreated cells of *S. aureus* had a distinct structure with cocci shape and intact cellular walls, and showed no signs of damage or morphological change (Figure 6A). Some deformations were observed in *S. aureus* treated by *M. pulegium* extract (Figure 6B,C), including partially digested cell wall and exfoliated fragments from the surface of bacterial cell, exposition of the patches in the cytoplasmic membrane, lack of cell wall or its detachment from the membrane, thickening of the outer membrane, and abnormal septa with the presence of ghost cells. Toxic effects of the *M. pulegium* extract on the structure and function of the membrane have generally been used to describe the antimicrobial action of the extract and its components. Untreated *P. aeruginosa* cells revealed a normal rod-shaped structure with an intact outer membrane and the cytoplasmic membrane near the cell wall, and the intracellular content was well-maintained. The periplasmic area was narrow and homogenous (Figure 6D). Several alterations in cell shape were detected after treatment with *M. pulegium* extract, and the periplasmic space indicated separation of the cell wall from the plasma membrane. The internal structures were disordered, and the components of the bacterial cell membrane were distorted and distributed from their original form. Draining of the intracellular contents was seen at particular sites within the cells, mostly at the septal and polar regions, due to breaches of the cell membrane. There were numerous cells without membranes and null cells of *P. aeruginosa* (Figure 6E,F). Approximately the same alteration was recorded in *P. aeruginosa*, leading to cell death [50].

### 2.3. Antioxidant Activity of M. pulegium Extract

The extract of *M. pulegium* showed excellent antioxidant activity according to the DPPH scavenging method. DPPH scavenging (%) increased with the increment of concentration, and reached 88.1% at 1000 μg/mL (Table 4). Surprisingly, the IC_50_ of *M. pulegium* extract was 18 μg/mL, compared with IC_50_ of 15.0 μg/mL for ascorbic acid. The current results can therefore confirm the antioxidant activity of this extract. The current study suggested that the main contributors to the antioxidant potential are associated with phenolic and flavonoid contents. Strong antioxidant activity of *M. pulegium* extract has been documented via numerous techniques including DPPH, trolox equivalent antioxidant capacity, ferric reducing antioxidant power, and oxygen radical absorbance capacity [47]. Many studies have reported relationships between phenolic content and antioxidative activity in plants [10,11]. However, the IC_50_ of the *M. pulegium* essential oil was 7.7 mg/mL [48] less than the current result (IC_50_, 18 μg/mL) for *M. pulegium* extract, although Aimad et al. [48] confirmed their obtained result using essential oil while our study dealt with the raw extract. Aimad et al. [48] suggested that the antioxidant activity of *M. pulegium* extract obtained by total antioxidant capacity and DPPH techniques may be due to the occurrence of pulegone.

### 2.4. Hemolysis Inhibition by M. pulegium Extract

According to the obtained results, all the applied concentrations (100–1000 μg/mL) of the *M. pulegium* extract inhibited the lysis of the erythrocyte membrane as shown (Table 5 and Figure 7). Hemolysis inhibition increased with increment of the concentration, and reached 79.8% at 1000 μg/mL (Table 5). The result was compared using 200 μg/mL of indomethacin, where hemolysis inhibition was 91.0%. Hemolysis inhibition reflects the anti-inflammatory activity of *M. pulegium* extract, and the obtained results are in accordance with previous studies demonstrating that *M. pulegium* exhibited anti-inflammatory activities [41,51]. There are similarities between the membranes of red blood cells and lysosomal membrane contents; lysis of lysosomes occurs during inflammation. When the red blood cells are exposed to hypotonic conditions, cell membrane lysis is caused, leading to hemolysis in addition to haemoglobin oxidation. Prevention of hypo-tonicity leading to lysis of the red blood cell membrane that was performed to observe anti-inflammatory activity. The obtained results suggest a potential anti-inflammatory action that is probably due to the presence of pulegone-1,2-epoxide in the extract of the *M. pulegium*. This was noted in the GC-MS analysis, and it was previously demonstrated that pulegone can inhibit inflammation mechanisms [51,52]. Recently, Baali et al. [47] mentioned the promising activity of *M. pulegium* extract as a wound healing agent and documented a scientific rationale that confirmed the traditional application of this plant.

Anticoagulant activity of *M. pulegium* extract was evaluated using APTT assay, measuring the activity of all coagulation factors in the intrinsic pathway, and PT assay, measuring the activity of the extrinsic pathway. PT and APTT results increased with incremental concentrations of the *M. pulegium* extract (Figure 8). *M. pulegium* extract showed PT and APTT of 21.5 s and 49.5 s, respectively (Figure 8), while heparin as a positive control showed PT and APTT as 115 s and 134.2 s, respectively, at 50 μg/mL (Figure 9). At 75 μg/mL of *M. pulegium* extract, PT and APTT were at 26.7 s and 67.8 s, respectively. Recently Leite et al. [53] studied the anticoagulant activity of numerous plants, among which *Mentha crispa* (APTT = 51.25) showed greatest anticoagulant potential. Ku and Bae [54] suggested that the inhibition of intrinsic and common pathways was associated with the prolongation of APTT, while the inhibition of the extrinsic coagulation pathway was associated with the prolongation of PT. The anticoagulant activities exhibited by *M. pulegium* extract may be due to the presence of natural constituents as observed by GC-MS analysis in the current study. The obtained results proved that the *M. pulegium* extract was effective as an anticoagulation agent.

### 2.5. Anticancer Activity of M. pulegium Extract

Cytotoxic activity of *M. pulegium* against two cancer cell lines (PC3 and MCF7) was assessed by MTT in vitro. The obtained results showed a dose-dependent decline in cancer cell line proliferation after exposure to *M. pulegium* extract. The highest cytotoxic activity was 97.17% and 96.71% against PC3 and MCF7, respectively, at 500 µg/mL (Table 6). PC3 was more resistant than MCF7 at low concentrations of *M. pulegium* extract, where acute toxicity was 0.60 and 10.34% for PC3 and MCF7, respectively, at 31.25 µg/mL. Toxicity was 28.88 and 48.46% for PC3 and MCF7, respectively, at 62.5 µg/mL. A similar study was conducted previously [55], where the anticancer potential of *M. pulegium* was documented against the MCF7 breast cancer cell line with acute toxicity of 88.547% at 500 μg/mL. However, the anticancer activity was recorded against the two tested cells, but different values of IC_50_ (97.99 and 80.21 µg/mL for PC3 and MCF7, respectively) were recorded according to the type of cell (Table 6) or plant origin, as reported in other literature. Low IC_50_ (59 and 48.86 µg/mL against MCF7 and A549, respectively) of *M. pulegium* L extract was recorded in another study [56]. Numerous authors have mentioned that the anti-proliferative and apoptotic activities of *M. pulegium* L extract, as well as those of other plants, may be due to the presence of flavanones, particularly hesperidin [10,20]. Cell death mechanism caused by the *M. pulegium* L extract was observed by cell morphology assay, where the acute toxicity was accompanied by a morphological change of PC3 and MCF7 (Figure 10), particularly at high concentrations. Cell shrinkage was observed with exposure to 125 µg/mL, in addition to the appearance of round cells and membrane breakdown with cell disruption at 250 and 500 µg/mL, possibly underlining a genotoxic interference, although this postulation requires further study.

### 2.6. Molecular Modeling: Docking Study

As described in the “Method” section, the current docking protocol was authenticated for all target receptors. Neophytadiene, luteolin, chrysoeriol, and kaempferol were analyzed for binding affinity with *P. aeruginosa* (PDB = 7BCZ), crystal structure of *E. coli* DNA (PDB = 7C7N), crystal structure of human prostate (PDB = 3QUM), and the crystal structure of the breast-cancer-associated protein (PDB = 1JNX) (Figure 11). The amino acid residue interactions of different proteins with active compounds are presented in Figure 12. Molecular modeling calculation was carried out to investigate the binding free energies of this inhibitor inside the target receptor, as mentioned previously [23].

Docking of neophytadiene with active receptor sites of (7BCZ) indicated the occurrence of a hydrogen bond among C1 atoms in the ligand with aromatic ring PHE191 amino acids residue at a distance of 4.37 ^o^A. Also, the interaction between luteolin and the active site binding of *E. coli* DNA (7C7N) revealed the presence of a hydrogen donor atom between the O28 atom in the ligand and ASP 45 amino acid residue, at a distance of 3.26 ^o^A. The docked chrysoeriol and kaempferol were superimposed on the crystal structure of human prostate (PDB = 3QUM) and the crystal structure of the breast-cancer-associated protein (PDB = 1JNX), with RMSD values (1.0946, 1.7265) and (1.7518, 1.3456), respectively, and binding free energies of (−6.3350 kcal·mol^−1^, −5.0644 kcal·mol^−1^) and (−6.3593 kcal·mol^−1^, −5.2916 kcal·mol^−1^), respectively. The total obtained results of docking scores and energies of compounds are presented in Table 7. The interactions between almost all atoms in the compounds and amino acid residues of enzymes are shown in Table 8. The effectiveness of the compounds tested in the current study was evaluated on the basis of docking scores, as described previously [24,25] for other compounds, namely deguelin and its derivatives.

## 3. Material and Methods

### 3.1. Collected Plant Samples and Process of Extraction

Leaves, stems, and flowers of *M. pulegium* were collected from Menofia Governorate throughout August 2021. The collected plant parts were washed with running tap water followed by sterile water to eliminate any dust, then dried and ground by electric mixer. Then, the obtained dried plants (140 g) were ground using the electric mixer into a fine powder, homogenized, then macerated in a stoppered container with 500 mL of 85% methanol for a period of 7 days. Then, for conventional extraction, the extract was processed in a sonicator at 40 °C for 60 min. This extract was filtered, then concentrated in a vacuum at 40 °C for 30 min using Rota vapor, to provide 7.0 g crude extract. Chemical analysis and biological activity analyses were performed on the extract.

### 3.2. Analysis of M. pulegium Extract by Gas Chromatography-Mass Spectrometry (GC-MS)

GC (THERMO Scientific Corp., Waltham, MA, USA)-MS (ISQ Single Quadrupole Mass Spectrometer) was used for *M. pulegium* extract analysis. Chromatographic conditions included film thickness of the capillary column TR-5MS of 30 m × 0.32 mm × 0.25 μm. Temperature cycling to 60 °C was used for the initial analysis, incrementing to 240 °C, followed by a gradual increase to the maximum temperature 290 °C by 30 °C/min, remaining isothermally constant up to 2 min. The protected temperature for injector was 250 °C and for MS transfer was 260 °C. One mL/min of high purity helium at constant flow was applied as carrier. One µL extract of *M. pulegium* was injected using the Auto sampler AS1300 linked to GC in the split manner. At a range of *m*/*z* 40–1000 achieved by electron energy with 70 eV application, the spectra of the electron ionization mass were collected in full scan mode. By calculation of mass spectra and retention time (RT) of the detected contents, these were identified by comparison with the existing evidence in the mass spectral library at the National Institute of Standards and Technology [1].

### 3.3. High-Performance Liquid Chromatography (HPLC) for Flavonoid and Phenolic Contents Determination

The analysis of flavonoid and phenolic contents of the extract was performed using HPLC. The chromatographic conditions included HPLC-(Agilent 1100) (Agilent, Santa Clara, CA, USA), equipped with two LC pumps, a UV/Vis detector, 125 mm × 4.60 mm, with 5 µm particle size of the C18 column. Twenty µL of the extract was injected in HPLC using methanol and 1/25 of acetic acid/water (60:40 *v*/*v*) as a mobile solvent for separation of phenolic acids. In case of flavonoids separation, the used mobile solvent with an isocratic flow rate of 1.0 mL min^−1^ was methanol/water in a 50:50 ratio *v*/*v*, adjusted using phosphoric acid to obtain pH 2.8. The obtained peaks of the separated flavonoid and phenolic content were compared with the different standard stock solutions of pure phenolic and flavonoids in methanol extract, using the RT (retention time) and UV-Vis spectra [57].

### 3.4. Antimicrobial Potential of M. pulegium Extract

Study of the antimicrobial potential of *M. pulegium* extract was carried out using well diffusion assay against bacteria including four bacterial species (*Bacillus subtilis*, *Pseudomonas aeruginosa*, *Staphylococcus aureus*, *Escherichia coli*), as well as yeast (*Candida albicans*), and filamentous black mold (*Mucor circinelloides*). Ain Shams University Hospitals, Egypt were the source of the tested bacteria, while Assiut University Mycological Centre provided the tested fungi. A hole (6 mm) was cut aseptically into agar using a sterile cork borer; extract solution (100 ul) was poured into the hole. Next, the plates were inoculated, then kept for 30 min in the refrigerator for proper diffusion of the extract, followed by incubation at 28 °C for fungi and 37 °C for bacteria and yeast. The diameter of the apparent inhibition zone (IZ) surrounding the wells was measured (mm) [58]. Gentamycin and ketoconazole were used as antibiotic and antifungal, respectively, as positive control. Dimethyl sulfoxide (DMSO) as a solvent of plant extract was applied as control. MIC was detected by microdilution protocol by making serial dilutions of 1 mL of *M. pulegium* extract (dissolved in DMSO) in broth, using a new pipette for each subsequent dilution step. The antimicrobial and broth solutions were dispensed into the plastic microdilution trays. The inoculum was prepared by making saline suspension of bacterial or yeast colonies (2 × 10^8^ colony-forming units/mL) from 20 h old, cultivated agar plates; 100 μL from the relevant suspension was inoculated into each well. Then the inoculated macro-dilution tubes were incubated (35 ± 2 °C) for 18 h (specific for bacteria) or for 24 h (specific for yeast) The MIC of *M. pulegium* extract was detected by measuring OD at 600 nm, and compared with growth without treatment as control [18].

### 3.5. Antioxidant Activity of M. pulegium Extract

The antioxidant activity of the extract was observed using a 2, 2-diphenyl-1-picrylhydrazyl (DPPH) radical scavenging test. Different concentrations of the dried methanolic extract of *M. pulegium* were dissolved in DMSO. Then the different concentrations (15–1000 μg/mL) were prepared by dilution, and 2 mL of 0.1 mM prepared DPPH in methanol as solvent solution was added to each of the diluted extracts, then mixed by vortex, followed by keeping under shade for 1 h. The addition of 2 mL of the DPPH solution to 1 mL of methanol was utilized as a negative control [59]. The mean of three absorbance readings of the plant extract and negative control at 514 nm was recorded, then the activity of DPPH radical scavenging (%) was determined by the following formulation:$$\mathrm{DPPH}\;\mathrm{scavenging}\;\mathrm{activity}\;\left(\%\right)=\frac{\mathrm{Control}\;\mathrm{sample}\;\mathrm{absorbance}\;-\mathrm{Extract}\;\mathrm{sample}\;\mathrm{absorbance}\;}{\mathrm{Control}\;\mathrm{sample}\;\mathrm{absorbance}}\times 100\;$$

Inhibitory concentration 50% (IC_50_) of the extract was determined.

Ascorbic acid at different amounts from 5 to 80 μg/mL was used to determine antioxidant activity as a positive control, as mentioned in the description of the antioxidant activity of the extract.

### 3.6. Anti-Hemolytic Activity Evaluation of M. pulegium Extract In Vitro

For the erythrocyte preparation, 3 mL of fresh whole blood was collected from one of the current research authors, in heparin blood collection tubes. The collected red blood cells from the blood sample obtained via centrifugation at 3000 rpm for 10 min were dissolved using normal saline solution (*v*/*v* supernatant), then reconstituted up to 40% *v*/*v* suspension by sodium phosphate (10 mM) as isotonic buffer solution adjusted to pH 7.4. Successive dilutions of plant extract in DMSO (from 100 to 1000 μg/mL) were added to 5 mL of distilled water to obtain the hypotonic solution.

Suspension of erythrocyte (0.1 mL) was added to the prepared hypotonic solution, followed by gentle mixing and incubation (37 °C) for 60 min. For the next incubation period it was centrifuged at 1200× *g* for 5 min, and the collected supernatant was subjected to assessment of its haemoglobin content via spectrophotometer (Milton Roy) at 540 nm. Indomethacin at 200 μg/mL was used as a positive control. Haemolysis inhibition (%) caused by the plant extract was calculated by the equation:$$\mathrm{Haemolysis}\;\mathrm{inhibition}\;\left(\%\right)=1-\frac{\;\mathrm{AD}2-\mathrm{AD}1}{\mathrm{AD}3-\mathrm{AD}1}\times 100$$
where AD1 is the absorbance of the treated sample in the isotonic solution, AD2 is the absorbance of the treated sample in the hypotonic solution, AD3 is absorbance of the control sample in the hypotonic solution. Hemolysis % was assessed with regard to hemolysis in distilled water (100%) [60].

### 3.7. Coagulate Activity Evaluation of M. pulegium Extract In Vitro

The dried methanolic extract of *M. pulegium* was dissolved in dimethyl sulfoxide (DMSO) then evaluated as an anticoagulant agent using prothrombin time (PT) and activated partial thromboplastin time (APTT) as a classical coagulant test. Nine parts of blood collected from an author of the current research were added to one part of 3.2% sodium citrate, followed by centrifugation (5000× *g* rpm) for 10 min, and the collected supernatant was utilized for carrying out the PT and APTT tests. Citrated normal human plasma was combined with different dilutions of the extract and incubated for 3 min at 37 °C. The reagents of the PT and APTT were incubated at 37 °C for 3 min before carrying out the tests, then 0.10 mL of the APTT reagent was added to each mixture of plasma and extract, then incubated at 37 °C for 5 min, then 0.10 mL of 0.025 mol/L CaCl_2_ (pre-incubated at 37 °C for 3 min) was added, and the clotting time was recorded. Mixtures of citrated normal human plasma with different dilutions of the extract were amended with 0.20 mL PT reagent, then clotting times were recorded to assess PT. These procedures were carried out using heparin as control [57].

### 3.8. Antitumor Assay and Morphological Characteristics of the Treated Cancer Cells

Cytotoxicity of the extract against MCF7 and PC3 cells was performed via 3-(4,5-dimethylthiazol-2-yl)-2,5-diphenyltetrazolium bromide (MTT) assay. The solution of MTT was prepared in phosphate buffered saline (PBS) as 5 mg/mL. The cells (1 × 10^5^ cells/mL) were inoculated in the 96-well tissue culture plate, followed by incubation at 37 °C for 24 h to develop a complete monolayer sheet. Then, the growth medium was decanted from the 96-well micro titer plates after confluent sheets of cells were formed, and the cell monolayer was washed twice with wash media. Six-fold dilutions of the extract were made in Roswell Park Memorial Institute (RPMI) 1640 medium with serum (2%) as maintenance medium. Each dilution (0.1 mL) was tested in different wells, leaving three wells as control receiving only maintenance medium, then incubated at 37 °C and scanned for any physical signs of toxicity. To each well, 20 µL MTT solution was added, and the MTT was mixed into the media. It was placed on a shaking desk for 5 min at around 150 rpm, then incubated at 37 °C under 5% CO_2_ for 1–5 h to permit the metabolization of MTT. The resulting MTT metabolic product was re-suspended in 200 µL of DMSO, followed by shaking at 150 rpm for 5 min to mix it into the solvent. The optical density (correlated directly with cell numbers) was measured at 560 nm with subtract background at 620 nm. Cell lines in growth medium without plant extract were studied as a control [59].

### 3.9. Molecular Docking

A molecular modeling study was undertaken using the Molecular Operating Environment (MOE) [61], a drug discovery software platform that integrates visualization, modeling and simulations, as well as methodology development, into one package. Molecular modeling studies were performed using Molecular Operating Environment (MOE, 2019) software on Dell Core i7 processor with 1.9 GH, 16 GB memory, and a Windows 10, 64-bit operating system. Energy minimizations were performed using MOE, with a RMSD (root mean square deviation) gradient of 0.05 kcal/mol and MMFF94X force field, and partial charges were automatically calculated. The X-ray crystallographic structures were obtained from the Protein Data Bank (PDB) (www.rcsb.org/pdb accessed on 19 July 2022); the PDB files were (7BCZ), (7C7N), (3QUM) and (1JNX).

The target proteins were prepared for docking by:Removing the water molecules and co-ligand from the active site of the protein.Addition of hydrogen atoms to the structure, with standard geometry.Using the MOE site finder to generate the active binding sites, to create the dummy sites as the binding pocket.Saving the obtained pocket was saved in MOE, to be used for predicting the ligand protein.

Moreover, the native ligands were minimized to their lowest energy using the MMFF94 force field. Then, the final form was obtained after 3D protonation and the correction process. The general docking scenario was run to 100_ns_ on the rigid receptor atoms, and the ligands were placed in the site using the triangle matcher method. London dG was utilized as a scoring function and the GBVI/WSA dG methods were used for rescoring. The five best poses were ranked by their binding free energy (S, kcal/mol), and the lengths of hydrogen bonds between the compounds and amino acids in the protein did not exceed 3.5 A°. RMSD and RMSD-refine fields were used for pose-with-pose comparison of the results in the co-crystal ligand position before and after amendment, respectively.

### 3.10. Statistical Analysis

The current experiments were realized in replicate, and results were therefore calculated as ± standard deviation (SD) or standard error (SE) means. GraphPad Prism^®^ (version 5.0, San Diego, CA, USA) software was applied to obtain graphs for the IC_50_ values of DPPH radical scavenging activity.

## 4. Conclusions

The obtained findings highlight the phytochemical and pharmaceutical application of *M. pulegium* extracts. In the current study, the extract was rich in several active components that showed biological activity including anticancer, antimicrobial, antioxidant, and as anti-hemolytic activity. These data suggest that this plant extract is a potential candidate for further experiments towards its use as an alternative drug. The MOE molecular modeling environment was used to study inhibitor activity for (7BCZ), (7C7N), (3QUM) and (1JNX). It was found that the energy scores of the molecular docking study were in good agreement with the experimental results. In conclusion, the antibacterial and antifungal activities of *M. pulegium* extract obtained from plants growing in Egypt were more effective than traditional antibiotics against the microorganisms studied.

## Figures and Tables

**Figure 1 molecules-27-04824-f001:**
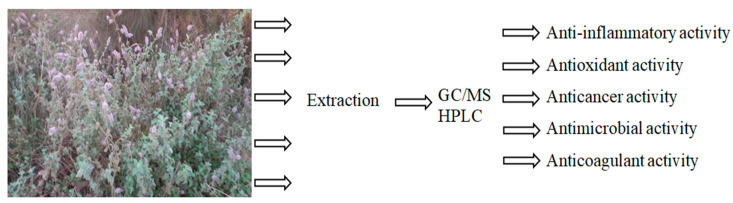
Diagram illustrating *M. pulegium* plants with the analysis methods and biological activities.

**Figure 2 molecules-27-04824-f002:**
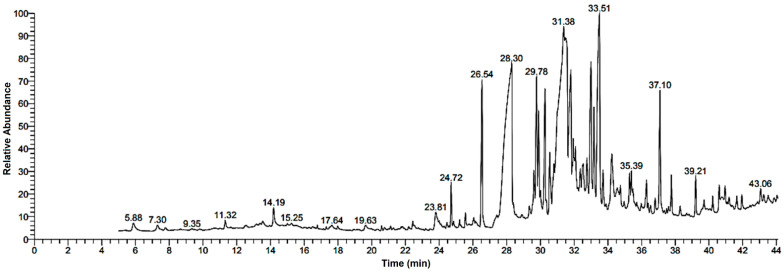
GC-MS chromatogram analysis of *M. pulegium* extract.

**Figure 3 molecules-27-04824-f003:**
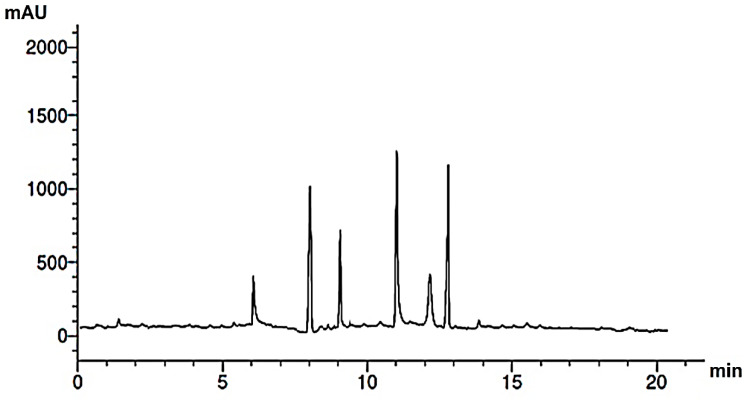
HPLC chromatogram of phenolic acids of *M. pulegium* extract.

**Figure 4 molecules-27-04824-f004:**
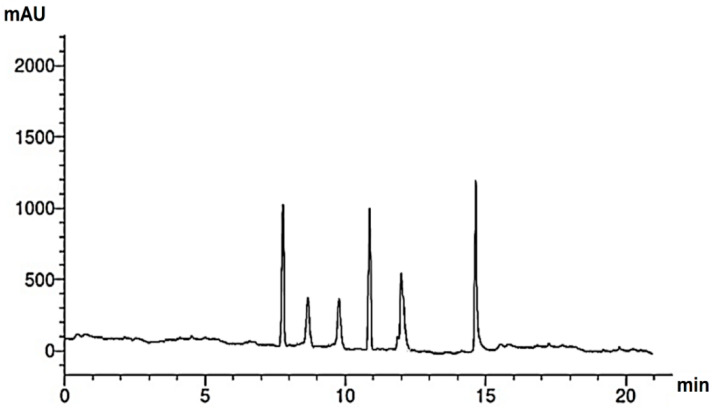
HPLC chromatogram of flavonoids of *M. pulegium* extract.

**Figure 5 molecules-27-04824-f005:**
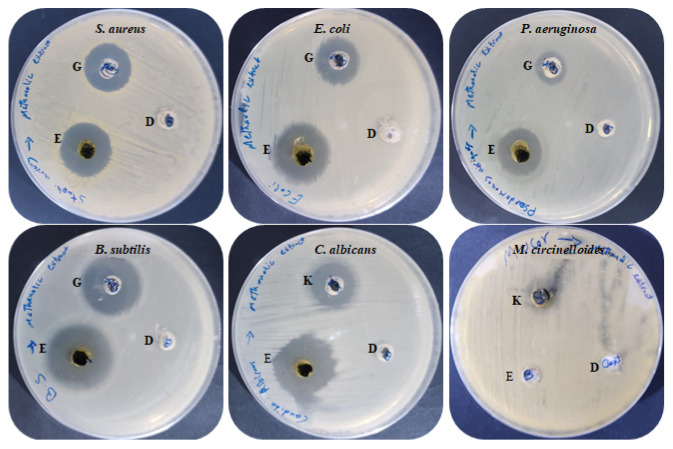
Antimicrobial activities of *M. pulegium* extract against different bacteria and fungi: (E) *M. pulegium* extract; (D) methanolic loaded disc, (G) Gentamycin, (K) Ketoconazole.

**Figure 6 molecules-27-04824-f006:**
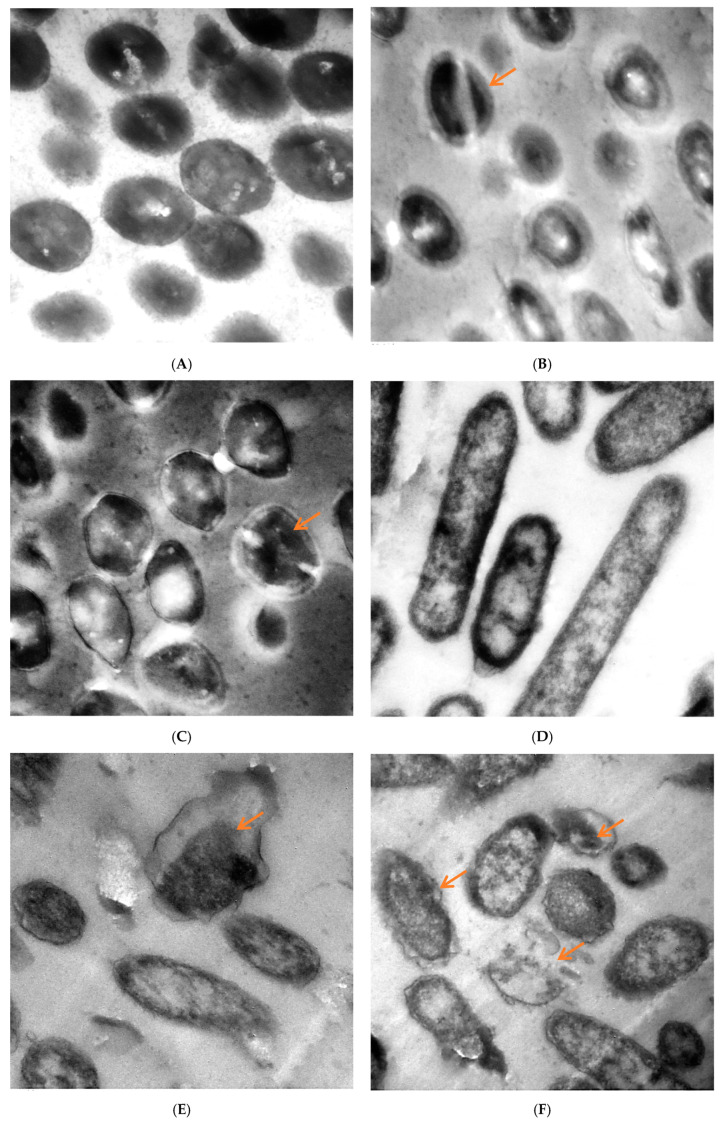
TEM micrographs of (**A**) untreated *S. aureus*, (**B**) S. *aureus* treated at 62.50 µg/mL and (**C**) at 100 µg/mL, (**D**) untreated *P. aeruginosa*, (**E**) *P. aeruginosa* treated with *M. pulegium* extract at 15.62 µg/mL and (**F**) at 30 µg/mL. Scale Bar = 100 nm, 5000×.

**Figure 7 molecules-27-04824-f007:**
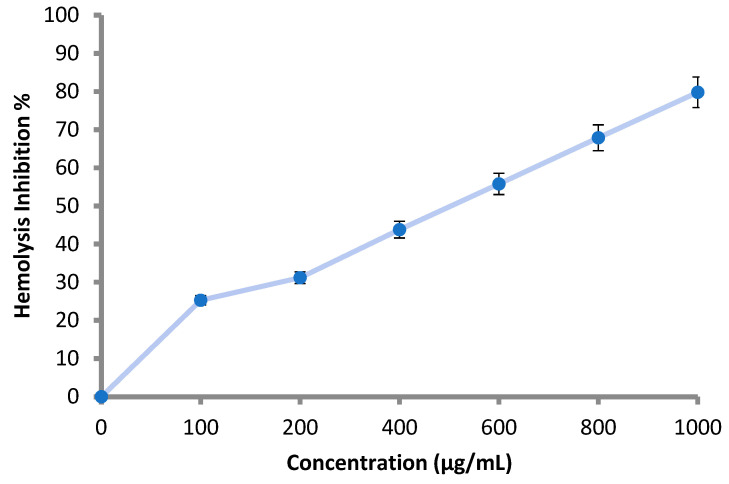
Hemolysis inhibition (%) at different concentration of *M. pulegium* extract 3.5. Anticoagulant Activity of *M. pulegium* Extract.

**Figure 8 molecules-27-04824-f008:**
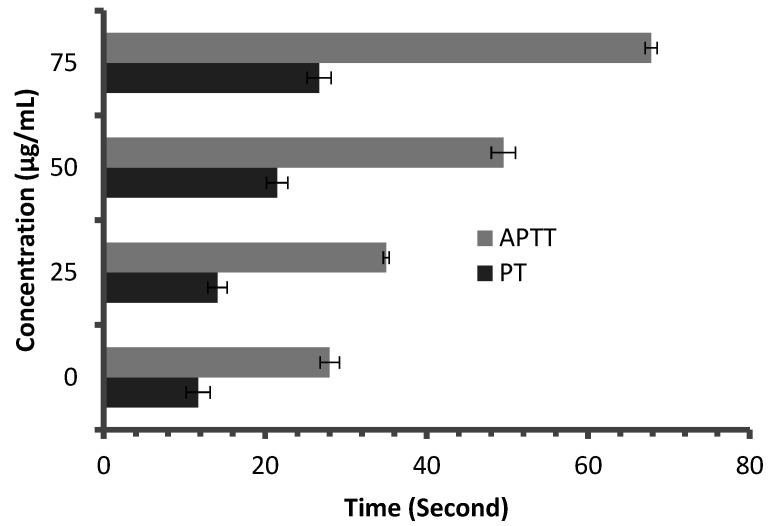
Anticoagulant activity of *M. pulegium* at different concentrations, represented by PT and APTT.

**Figure 9 molecules-27-04824-f009:**
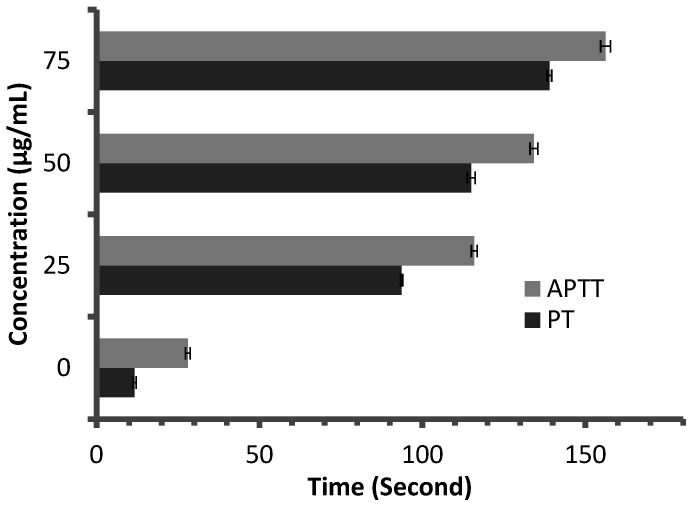
Anticoagulant activity of heparin at different concentrations, represented by PT and APTT.

**Figure 10 molecules-27-04824-f010:**
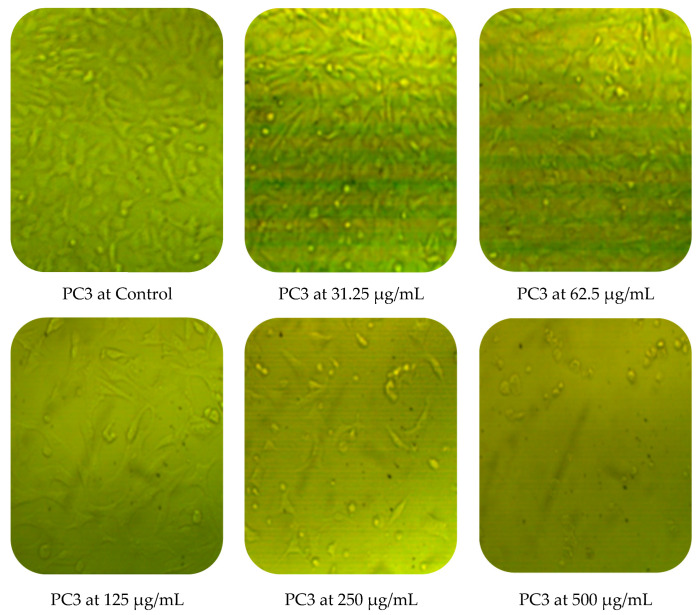
Effects of different concentrations of *M. pulegium* extract on morphological changes of PC3 and MCF7.

**Figure 11 molecules-27-04824-f011:**
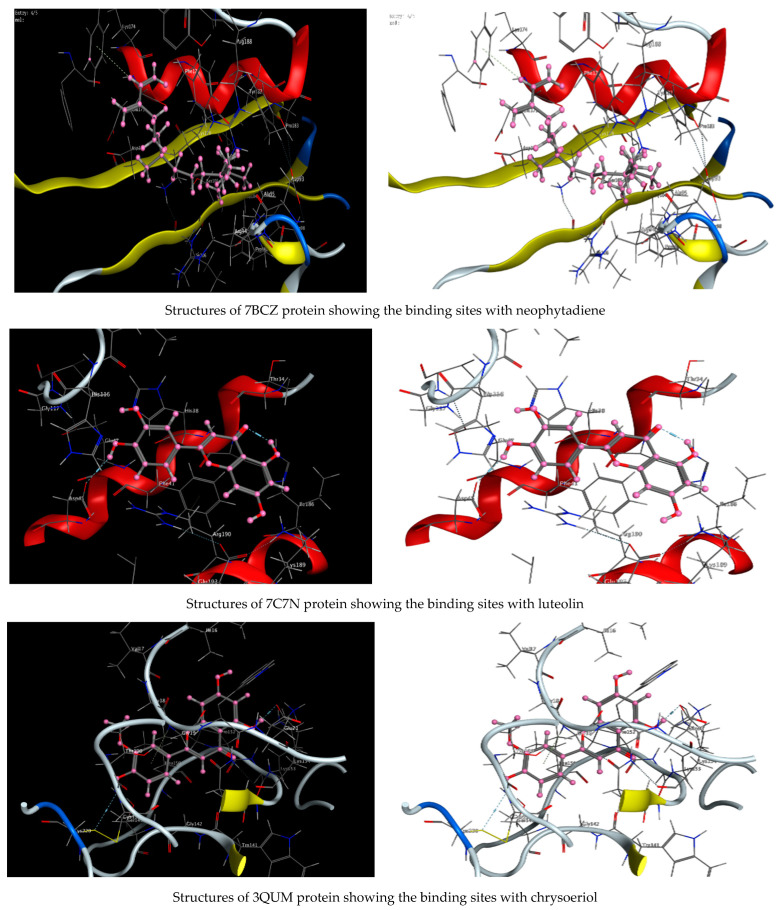
Structures of different proteins showing their respective binding sites for active compounds.

**Figure 12 molecules-27-04824-f012:**
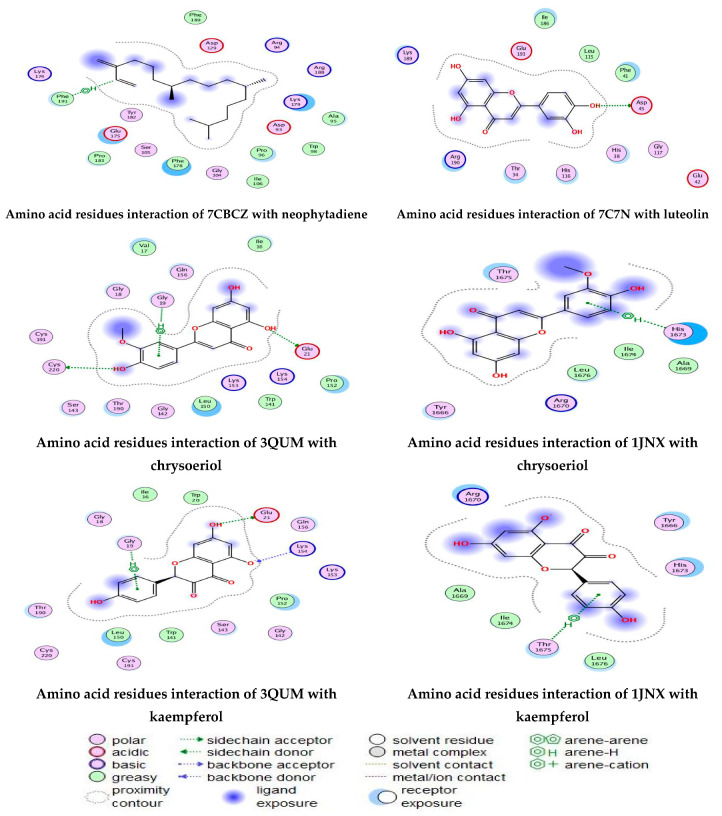
Amino acid residue interactions of different proteins with active compounds, and the representative key of the types of interaction between active compounds and proteins.

**Table 1 molecules-27-04824-t001:** Phyto-constituents of *M. pulegium* extract identified by GC-MS.

Phyto-Constituent	RT *	Area %	Molecular Formula	Molecular Weight
Pulegone-1,2-epoxide (oxygenated terpene)	14.18	0.52	C_10_H_14_O_2_	166
Methyl tridecanoate	23.80	0.69	C_14_H_28_O_2_	228
Neophytadiene (Diterpenoid)	24.72	1.07	C_20_H_38_	278
2-cis-9-Octadecenyloxyethano	25.56	0.36	C_20_H_40_O_2_	312
Methyl hexadecanoate	26.54	4.56	C_17_H_34_O_2_	270
Hexadecanoic acid	28.30	20.88	C_16_H_32_O_2_	256
2,3-Dihydroxypropyl palmi	29.34	0.32	C_19_H_38_O_4_	330
Linolelaidic acid, methyl ester	29.62	0.87	C_19_H_34_O_2_	294
Elaidic acid, methyl ester	29.78	4.17	C_19_H_36_O_2_	296
cis-11-Octadecenoic acid, methyl ester	29.90	2.80	C_19_H_36_O_2_	296
Isochiapin B	30.01	0.32	C_19_H_22_O_6_	346
Octadecanoic acid, methyl ester	30.27	3.94	C_19_H_38_O_2_	298
Methyl (7E,10E)-7,10-octadecadienoate	30.55	2.06	C_19_H_34_O_2_	294
Ethyl linoleate	30.80	0.65	C_21_H_38_O_2_	322
Oleic acid	30.99	0.79	C_18_H_34_O_2_	282
6-Octadecenoic acid	31.39	3.58	C_18_H_34_O_2_	282
cis-13-Octadecenoic acid	31.57	6.84	C_18_H_34_O_2_	282
Stearic acid	31.81	4.87	C_18_H_36_O_2_	284
19-nor-4-androstenediol (Phytosterol)	31.94	1.39	C_18_H_28_O_2_	276
Androstan-17-one (Phytosterol)	32.08	1.20	C_21_H_34_O_2_	318
cis-13-Eicosenoic acid	32.36	0.52	C_20_H_38_O_2_	310
Oxiraneoctanoic acid, 3-octyl-methyl ester	33.00	4.72	C_19_H_36_O_3_	312
Methyl 10-ketostearate	33.50	10.84	C_19_H_36_O_3_	312
Arachidic acid methyl ester	33.71	1.00	C_21_H_42_O_2_	326
7,7,8,8,9,9,10,10,10-Nonafluorodecane-1,2-Diol	34.24	2.10	C_10_H_13_F_9_O_2_	336
Dasycarpidan-1-methanol, acetate (ester)	34.56	0.75	C_20_H_26_N_2_O_2_	326
6,7-Dimethoxy-1,2-Dimethyl-1,2,3,4-Tetrahydro-5-isoquinolinol	35.39	0.95	C_13_H_19_NO_3_	237
Hahnfett	35.48	0.34	ND **	ND **
2,3-Dihydroxypropyl elaidate	35.29	0.79	C_21_H_40_O_4_	356
Dotriacontane	36.29	0.84	C_32_H_66_	450
Di-2-Benzothiazole Disulfane	36.80	0.36	C_14_H_8_N_2_S_4_	332
1,2-Benzenedicarboxylic acid	37.09	3.60	C_24_H_38_O_4_	390
Heptacosane	37.78	0.96	C_27_H_56_	380
Docosanoic acid, 1,2,3-propanetriylester	38.29	0.31	C_69_H_134_O_6_	1058
Docosanoic acid, methylester	39.71	0.49	C_23_H_46_O_2_	354
Cholesterol margarate(Phytosterol)	41.65	0.42	C_44_H_78_O_2_	638
6,8-DI-C-á-Glucosylluteolin	41.95	0.39	C_27_H_30_O_16_	610
Stigmast-5-EN-3-OL (Phytosterol)	43.06	0.48	C_28_H_44_O_4_	444

* RT, retention time; ** ND, not detected.

**Table 2 molecules-27-04824-t002:** Flavonoids and phenolic acids contained in *M. pulegium* extract, identified by HPLC.

Flavonoids			Phenolic Acids		
RT *	Compound	Concentration (µg/mL)	RT *	Compound	Concentration (µg/mL)
7.8	Kaempferol	11.14	6	*p*-Coumaric acid	5.06
8.7	Luteolin	2.36	8	Caffeic acid	11.45
9.8	Hesperidin	3.05	9	Pyrogallol	9.36
11	7-OH flavone	10.14	11	Ferulic acid	13.19
12	Catechin	4.11	12	Salicylic acid	4.17
15	Chrysoeriol	15.36	13	Cinnamic acid	12.69

* RT, retention time.

**Table 3 molecules-27-04824-t003:** Antimicrobial activity of *M. pulegium* extract.

Tested Microorganisms		Inhibition Zone (mm)		
		Extract (100 µL)	Control *	MIC *** µg/mL
Gram-positive bacteria	*S. aureus*	23	21	62.50
	*B. subtilis*	27	25	45.25
Gram-negative bacteria	*E. coli*	26	20	33.60
	*P. aeruginosa*	19	14	15.62
Fungi	*C. albicans*	25	22	65.20
	*M. circinelloides*	0.0	11	ND **

* Ketoconazole, or * Gentamycin as antifungal or antibiotic, respectively. ND **, not detected. ******* Minimum inhibitory concentration.

**Table 4 molecules-27-04824-t004:** Antioxidant activity of *M. pulegium* extract.

Concentration (μg/mL)	O.D * Mean	DPPH Scavenging (%)	SD **	SE ***
1000	0.172	88.1	0.003	0.001
500	0.215	85.1	0.004	0.001
250	0.307	78.7	0.002	0.001
125	0.419	70.9	0.003	0.001
62.5	0.529	63.3	0.003	0.001
31.25	0.653	54.7	0.005	0.001
15.62	0.762	47.1	0.006	0.002
**IC**_50_ of *M. pulegium* extract	18 μg/mL			
**IC**_50_ of ascorbic acid	15.0 μg/mL			

* OD, optical density; ** SD, standard deviation; *** SE, standard error.

**Table 5 molecules-27-04824-t005:** Effect of *M. pulegium* extract on hemolysis inhibition.

Concentration (μg/mL)	Hypotonic O.D *	Hemolysis Inhibition %	SD **	SE ***
Control	1.038	0	0.003	0.001
1000	0.301	79.8	0.009	0.003
800	0.417	67.9	0.005	0.002
600	0.527	55.8	0.008	0.003
400	0.641	43.8	0.002	0.001
200	0.766	31.2	0.004	0.001
100	0.822	25.3	0.008	0.002
Indomethacin at 200 μg/mL	0.130	91.0	0.005	0.002

* OD, optical density; ** SD, standard deviation; *** SE, standard error.

**Table 6 molecules-27-04824-t006:** Cytotoxicity of *M. pulegium* L extract against PC3 and MCF7 at different concentrations.

Concentration µg/mL	PC3				MCF7			
	Mean O.D *	SE **	Viability %	Toxicity %	Mean O.D *	SE **	Viability %	Toxicity %
Control	0.554	0.010	100	0.0	0.476	0.01	100	0.0
31.25	0.55	0.007	99.40	0.60	0.43	0.005	89.57	10.43
62.5	0.39	0.008	71.12	28.88	0.25	0.016	51.547	48.46
125	0.18	0.007	32.31	67.69	0.11	0.010	23.53	76.47
250	0.11	0.005	19.19	80.81	0.05	0.007	10.85	89.15
500	0.06	0.001	2.83	97.17	0.02	0.001	3.29	96.71
1000	0.02	0.001	3.13	96.87	0.02	0.001	3.71	96.29
IC_50_	97.99 µg/mL				80.21 µg/mL			

* OD, optical density; ** SE, standard error.

**Table 7 molecules-27-04824-t007:** Docking scores and energies of compounds with 7BCZ, 7C7N, 3QUM, and 1JNX proteins.

Compound	Receptor	mseq	S	rmsd_refne	E_conf	E_place	E_score1	E_refne	E_score2
Neophytadiene	7BCZ	1	−7.1580	1.2822	24.5600	−57.8312	−7.8518	−31.7035	−7.1580
Neophytadiene	7BCZ	1	−6.9125	2.3982	13.1121	−49.8856	−8.0566	−31.8298	−6.9125
Neophytadiene	7BCZ	1	−6.8013	1.3756	26.6025	−54.2040	−7.6831	−27.2223	−6.8013
Neophytadiene	7BCZ	1	−6.7640	1.1550	86.2092	−72.2128	−8.5950	−31.8384	−6.7640
Neophytadiene	7BCZ	1	−6.6269	1.8035	10.0713	−39.6643	−7.4382	−32.7957	−6.6269
Luteolin	7C7N	1	−5.2683	1.2104	−30.1027	−53.0515	−10.7811	−21.5450	−5.2683
Luteolin	7C7N	1	−5.2098	1.0321	−33.5362	−58.5419	−10.8163	−25.1026	−5.2098
Luteolin	7C7N	1	−5.1764	2.0993	−31.8830	−50.5262	−9.9086	−23.7861	−5.1764
Luteolin	7C7N	1	−5.0736	1.1214	−34.9514	−74.7281	−10.4863	−24.8905	−5.0736
Luteolin	7C7N	1	−5.0535	0.6345	−30.7855	−62.0186	−9.5565	−20.9379	−5.0535
Chrysoeriol	3QUM	1	−6.3350	1.0946	−15.0487	−75.6312	−11.4227	−36.8350	−6.3350
Chrysoeriol	3QUM	1	−6.2655	1.7381	−20.6640	−84.1280	−11.5780	−37.1195	−6.2655
Chrysoeriol	3QUM	1	−6.2171	0.7234	−14.4679	−76.0750	−11.4356	−33.1380	−6.2171
Chrysoeriol	3QUM	1	−6.1692	2.1441	−15.4071	−78.4542	−11.5705	−36.1058	−6.1692
Chrysoeriol	3QUM	1	−6.1571	1.1044	−15.2604	−84.8887	−11.8802	−35.8331	−6.1571
Chrysoeriol	1JNX	1	−5.0644	1.7265	−17.5303	−62.1961	−9.9634	−25.0239	−5.0644
Chrysoeriol	1JNX	1	−5.0101	0.9227	−15.7642	−42.2704	−9.1471	−24.0765	−5.0101
Chrysoeriol	1JNX	1	−4.9971	1.6559	−14.5476	−43.9988	−9.2024	−23.3186	−4.9971
Chrysoeriol	1JNX	1	−4.9616	1.4387	−16.8284	−44.6490	−9.6921	−23.2991	−4.9616
Chrysoeriol	1JNX	1	−4.8836	2.0835	−17.2031	−52.3723	−9.3292	−24.7783	−4.8836
Kaempferol	3QUM	1	−6.3593	1.7518	−54.3337	−63.8852	−12.2329	−35.5105	−6.3593
Kaempferol	3QUM	1	−6.2456	1.7957	−53.7207	−84.7006	−11.2066	−33.4084	−6.2456
Kaempferol	3QUM	1	−5.9016	1.4734	−50.9818	−65.7556	−11.2285	−27.7261	−5.9016
Kaempferol	3QUM	1	−5.8823	1.1136	−48.9402	−68.5376	−11.2814	−31.7357	−5.8823
Kaempferol	3QUM	1	−5.8293	1.7911	−50.1359	−67.1079	−11.2273	−31.8993	−5.8293
Kaempferol	1JNX	1	−5.2916	3.4082	−57.8566	−54.0164	−7.9325	−25.2956	−5.2916
Kaempferol	1JNX	1	−5.2264	1.3456	−58.4060	−23.5095	−7.0560	−24.2946	−5.2264
Kaempferol	1JNX	1	−4.9339	2.1388	−58.0519	−35.5016	−6.9689	−23.3973	−4.9339
Kaempferol	1JNX	1	−4.9124	2.8442	−58.3635	−31.1941	−7.3013	−22.9019	−4.9124
Kaempferol	1JNX	1	−4.8548	1.5679	−59.0462	−43.3097	−8.0880	−23.1065	−4.8548

Where S = final score, which is the score of the last stage that was not set to none. rmsd = root mean square deviation of the pose, in Å, from the original ligand. This field was exist if the site definition was identical to the definition of ligand. rmsd_refine = root mean square deviation between the pose before refinement and the pose after refinement. E_conf = energy of the conformer. If there was a refinement stage, this is the energy calculated at the end of the refinement. Note that for force field refinement, by default, this energy was calculated with the solvation option set to Born. E_place = score from the placement stage. E_score 1, E_score 2 = scores from rescoring stages 1 and 2. E_refine = score from the refinement stage, calculated to be the sum of the van der Waals electrostatic and solvation energies, under the generalized Born solvation model (GB/VI).

**Table 8 molecules-27-04824-t008:** Interaction of active compounds with proteins.

**Neophytadiene Interaction with 7BCZ Protein**				
**Neophytadiene**	**Receptor**	**Interaction**	**Distance**	**E (kcal/mol)**
C 1	6-ring PHE 191 (A)	Pi-H	4.73	−0.5
**Luteolin interaction with 7C7N protein**				
**Luteolin**	**Receptor**	**Interaction**	**Distance**	**E (kcal/mol)**
O 28	OD2 ASP 45 (A)	H-donor	3.26	−1.7
**Chrysoeriol interaction with 3QUM protein**				
**Chrysoeriol**	**Receptor**	**Interaction**	**Distance**	**E (kcal/mol)**
O 23	SG CYS 220 (P)	H-donor	4.29	−1.0
O 26	OE2 GLU 21 (P)	H-donor	2.85	−2.0
6-ring	N GLY 19 (P)	Pi-H	4.40	−0.6
**Chrysoeriol interaction with 1JNX protein**				
**Chrysoeriol**	**Receptor**	**Interaction**	**Distance**	**E (kcal/mol)**
6-ring	CB HIS 1673 (X)	Pi-H	3.65	−0.5
**Kaempferol Interaction with 3QUM protein**				
**Kaempferol**	**Receptor**	**Interaction**	**Distance**	**E (kcal/mol)**
O 28	OE2 GLU 21 (P)	H-donor	3.26	−0.9
O 27	N LYS 154 (P)	H-acceptor	3.01	−3.1
O 27	CE LYS 154 (P)	H-acceptor	3.18	−2.6
6-ring	N GLY 19 (P)	Pi-H	4.38	−1.5
**Kaempferol interaction with 1JNX protein**				
**Kaempferol**	**Receptor**	**Interaction**	**Distance**	**E (kcal/mol)**
6-ring	CA THR 1675 (X)	Pi-H	4.09	−1.1

## Data Availability

All data that support the findings of this study are available within the article.

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
