# Peer review of "Molecular Interaction Studies and Phytochemical Characterization of Mentha pulegium L. Constituents with Multiple Biological Utilities as Antioxidant, Antimicrobial, Anticancer and Anti-Hemolytic Agents"

_molecules, 2022, doi:10.3390/molecules27154824_

Round 1

Reviewer 1 Report

The text should be condensed during editing of language. Data such as HPLC profiles, Fig 10 and Tables 4 and 5 should be placed under supplementary data.

The MTT treatment time of 1-5 hr can only indicate immediate, acute toxicity, not clonogenic survival required for consideration as anticancer agents. Therefore all mention of cytotoxicity in results must be described as "acute toxicity".

Author Response

Very thanks for reviewing my manuscript 

1- Introduction: the following was added: 

Existence of several phenolic acids in M. pulegium like syringic acid, ferulic acid and as well as several flavonoid compounds, like isorhamnetin-3-O-glucoside and kaempferol-3-O-rutinoside; the presence of these compounds reflected high antioxidant activity of M. pulegium [12].

Alharbi et al.[11] studied the relationship among the biological utilities of two Mentha species (M. rotundifolia and M. pulegium) and its contents of phenolic compounds. Alharbi et al.[11] observed a positive correlation among phenolic contents and biological activities namely antimicrobial and antioxidant activities.

2- The text should be condensed during editing of language. Data such as HPLC profiles, Fig 10 and Tables 4 and 5 should be placed under supplementary data.

Response:  It's corrected, Fig. 10 and tables: Journal style transfer it to supplementary data

3- The MTT treatment time of 1-5 hr can only indicate immediate, acute toxicity, not clonogenic survival required for consideration as anticancer agents. Therefore all mention of cytotoxicity in results must be described as "acute toxicity".

Response : It changed to acute toxicity in the abstract and the results 

Reviewer 2 Report

Comments:

1- According the journal style: Results and Discussion section then Material and methods not the converse.

2- In lines 324 and 326, "HPLC chromatogram of" are bold. Change them.

3- Lines 377 and 378, it is okay but the font size of them are different. Please unify them.

4- Figure 10 contains some errors in typing of concentrations. Please revise it. It was in a correct structure in the previous version of the manuscript.

5- Some references are not unified with the others for example 11, 12, 19, 43 and 50. Please revise the whole references.

Author Response

Very thanks, my prof.

1- According the journal style: Results and Discussion section then Material and methods not the converse.

Response: In the first submission, I found  some papers result and discussion after  material and methods it's easy to transfer if the paper accepted 

2-  In lines 324 and 326, "HPLC chromatogram of" are bold. Change them.

Response: its changed 

3- Figure 10 contains some errors in typing of concentrations. Please revise it. It was in a correct structure in the previous version of the manuscript.

Responce : in the first submission it was correct but when the text transferred to Journal template it become incorrecte : in the final version it will corrected 

4. Some references are not unified with the others for example 11, 12, 19, 43 and 50. Please revise the whole references.

Response: all references will unified according to journal style  if paper accepted   

Reviewer 3 Report

The paper has a positive expression. However, the authors must work with the text and antioxidant activity (experimental studies)

1. Line 45. This sentence should be corrected like this. These compounds could be useful for the development of new therapeutic agents for cancer and infection treatment.

We do not treat bacteria

2. I recommend to give phrase the family Lamiaceae, not the Lamiaceae family

like this, The family Lamiaceae is in the major group Angiosperms (Flowering plants) like it is provided in Plant list

3. Lines 62-63 need an appropriate reference. The authors can take from the list of their references 

4. Line 63. ......the genus Mentha....

5. Line 67  -  reference is needed. Line 67-68 - this sentence is not clear absolutely.

6. Line 103 August 2021

7. Line 104-107. There should be a correction like this

and grinded via electric mixer. Then, the obtained dried plants (140g) were ground via the electric mixer to a fine powder, homogenized then macerated

in a stoppered container with 500 mL of 85% methanol for  a period of 7 days

8. Line 106-108. It is not clear when the author heated this extract

9. Line 137 and 96, probably for studying not studding

10. Line 126 - The analysis...

11. Line 160-162 The correction must be. Later. It is necessary to indicate the volume of the dilutions, probably 1.0 ml

12.  170-172 lines. There must be a correction. Ascorbic acid at different concentrations from 5 to 80 μg/mL was used to determine its antioxidant activity as a positive control as mentioned in the antioxidant activity of the extract

13. Chapter 2.6 needs a total stylistic and sense correction, especially Five mL of each distilled water. It is necessary to specify more clearly what is a control 

14. Conclusions. Please, remind about anti hemolytic activity

14. I strongly recommend to provide the structures of the main identified compounds for a better understanding of the text, especially those not typical for all plants, like Chrysoeriol, pulegone-1,2-epoxide

15. Pyrogallol is exactly not phenolic acid. Check also salicylic acid if it belongs to phenolic acids

16. Line 336 - correction

17. Line 349-350 need a correction is 

18. Chapter 3.3. It is recommended to widen, comparing to references

19. Which absorbance of the negative control was & I mean line 164

The addition of 2 mL of DPPH solution to 1 mL of methanol was used as a negative control [23]. As a rule absorbance of solutions should not exceed 0.8.  From table 4 It is seen, that the absorbance in some cases significantly exceeds the permitted value. So, it needs correction in the direction of performing experimental studies

20. Line 407. Something is missed There are similar.......

Some corrections are in the attached text

Author Response

All comments were corrected and changed with blue highlight

Round 2

Reviewer 1 Report

None

Author Response

Very thanks 

the results clearly  presented 

Author Response

Dear reviewer thanks for the fine review of my manuscript 

All request correction was done with a green highlight

the sentence was changed to:   The existence of several phenolic acids in M. pulegium like syringic acid, ferulic acid, and as well as several flavonoid compounds, like isorhamnetin-3-O-glucoside and kaempferol-3-O-rutinoside were reported; these compounds reflected high antioxidant activity [12]. A positive correlation among phenolic contents of two Mentha species (M. rotundifolia and M. pulegium) and its biological activities namely antimicrobial and antioxidant activities were documented by Alharbi et al.[11].

Table 4: the concentrations up to 15 μg/mL were deleted 

The obtained findings are highlighted the phytochemical and pharmaceutical application of M. pulegium extracts. In the current study, the extract is rich in several active components which showed biological activity such as anticancer, antimicrobial and antioxidant as well as anti-hemolytic activity. These data suggest that plant extract is a potential candidate for further experiments to be used as an alternative drug. MOE (molecular modeling environment) was performed as an inhibitor for (7BCZ), (7C7N), (3QUM) and (1JNX). It was found that the energy scores of the molecular docking study in good agreement with the experimental results. In conclusion, the antibacterial and antifungal activities of M. pulegium extracts obtained from plants growing in Egypt were more effective than traditional antibiotics to combat the microorganisms studied.

This manuscript is a resubmission of an earlier submission. The following is a list of the peer review reports and author responses from that submission.

Round 1

Reviewer 1 Report

  1. Better to say The family Lamiaceae according to The plant list http://www.theplantlist.org/browse/A/Lamiaceae/
  2. I wish the authors would remind by one sentence the information that flavons are typical flavonoids for the Lamiaceae family.
  3. Safflower oil ..... Is this component of the tested extract?
  4. Fig 5 - please correct the correspondence between the cups and letters. Any cup is not signed by a letter.
  5. 45 line - therapeutic drugs - better to change into herbal preparations,  line 44-49 - this sentence is very hard for understanding, 58-59 lines - four Gr+ve 58 and five Gr-ve bacteria - better to write the full forms like Gram-positive bacteria; lines 64-66 paraphrase this sentence, line 74 - please correct, line 79 - please, correct the sentence because it is understandable, line 93 - the protection; lines 111-114 - please, correct the sentence because it is impossible to understand; 104-105 lines -..... is rich in biological active constituents; 115 - ....identified IN the..... 166 line - ...all THe tested...220-221 lines - Therefore, the current result can indicate
     that this extract has antioxidant activity.
  6. 3.6 - add the information about the calibration curve of ascorbic acid there is no mention about ascorbic acid as a positive control
  7. Conclusions. This sentence is recommended to write like this - In the current study, the extract is rich in several active components which
    showed biological activity such as anticancer, antimicrobial, and antioxidant. In the current study, the extract exhibited anti-proliferative, antioxidant, antibacterial, and antiviral activity.

Reviewer 2 Report

There have been many reports about phytochemicals in penny royal, with  pulegone stated as the main active ingredient (primarily toxic)  and a wide variety of biological activities attributed to the plant. The content of specific compounds can vary widely depending upon the growth location, eg, Caputo et al (https://pubmed.ncbi.nlm.nih.gov/33918091/). The Introduction for this MS should cite Caputo et al or similar, to clarify the possibility of geographical variation in phytochemical content and, in view of its GRAS status, the potential toxicity of the plant notably with respect to pulegone content.  The apparent absence of pulegone and low level of its epoxide found in the present work may indicate that this source is safe for community consumption, a significant finding.

No new compounds appear to have been discovered here but the authors should indicate which of the known compounds that they list had not previously been reported in this plant.  

A considerable amount of data is given about the phytochemical contents and different in vitro bioactivities of the crude extract. The 80% methanol extraction would give a mixture of both hydrophobic and water-soluble compounds. Use of GC and HPLC respectively was appropriate. It is surprising in view of previous reports that more monoterpenes were not found. The solvent used to solubilize the extract should be stated for each bioassay. This affects the question of whether solubilization was adequate and the type of diluent appropriatefor a particular assay.

For the antitumor assay no treatment time was given. It is important with MTS and other growth assays that sufficient time is allowed for 3-4 rounds of replication; which in turn means that the initial cell number has to be <10,000/well and a treatment time of 5-7 days. A more fundamental problem here is that no normal cell was used for comparison with MCF7 and PC3.  I am unable to comment on the docking study.

Overall, the work at this stage, limited to a crude plant extract, does not advance knowledge much about likely uses for the plant extract in vivo.  Using this data as a starting point several of the identified compounds, selected for some unique feature (high %, not previously tested), should be tested in the appropriate bioassay. They could also be profiled in the docking study.  The results could then form the basis for a molecular profile that would guide selection of the species from different locations and justify treatment with one or more purified components.

Reviewer 3 Report

The manuscript entitled “Molecular Docking Studies and Phytochemical Characteriza- 2 tion of Mentha pulegium L. as Antimicrobial, Anticancer and 3 Anti-hemolytic Agent” cannot be accepted in its present form. My comments are as follows, The abstract is too lengthy. I would like the authors to modify the abstract part. I would suggest the authors to modify the sentences (54-55) “Numerous uses as drinking and food”. I would suggest the authors to check and rephrase the line 167 “Growth of B. subtilis (IZ, 27 mm) was more effected” effected? I would recommend the authors to italicize the appropriate words in line number 169-170 I would suggest to recheck the following sentences (For instance, line number 169-170; 196-197; 201-202; 220-228; 245-247 and many) In line number 198-199, the words used in the sentences found to be quite confusing and I would suggest the sentences to rephrase The figure caption (Fig 5) is quite confusing “Antimicrobial activities of M. pulegium extract against different bacteria and fungi S. aureus (A)” I would recommend the authors to mention the key findings using light color arrows and I haven’t find any scale bar in TEM images. I would suggest the authors to follow uniform space between the words throughout the manuscript. In line number 220 the terminology mentioned as 18 uμ/ml is quite confusing. Check and change as per the requirements I would suggest the authors to mention ml or mL throughout the manuscript. In line number 261 the authors have mentioned as “constituents as mentioned by GC/MS analysis” or “constituents as mentioned in GC/MS analysis” rephrase the word. The figure 9 was not clear and I would suggest the authors to replace with new images. Because, readers will not understood what exactly the figure conveys? The usage of singular and plural words followed unnecessary usage of capital letters and inappropriate usage of punctuations should be minimized. Overall, I would suggest the authors to seek native English speaker to improve the manuscript quality.

Reviewer 4 Report

The whole article needs a major reversion in organizing of presentation, spelling and typing.

Some comments:

  1. The introduction is not oriented into the target study in this paper, and there is any statement about the molecular docking.
  2. Line 108, abundant one15.36 μg/ml ---> abundant one (15.36 μg/ml)
  3. Some titles of figs. and tables are not bold and some of them are bold.
  4. Line 150 and 152, chromatograms -----> chromatogram
  5. Remove line 156.
  6. The (*) of RT* in table 2 indicates to what?
  7. Line 163, 2.1. Antimicrobial ----> 2.2. Antimicrobial
  8. Lines 169-171, extra words are italic.
  9. Line 158, check the number of figs.
  10. lines 207 and 208, remove S. aureus (A), P. aeruginosa (B), E. coli (C), B. subtilis(D), M. circinelloides (E) and C. albicans(F); L, latex; M, methanolic disc; G, Gentamycin; K, Ketoconazole
  11. Line 231 and 266, fig. (6) ----> fig. (7)
  12. Edit the number of all the incoming figs.
  13. Line 268, Mentha pulegium ----> M. pulegium
  14. Line 298, MSF7 ----> MCF7
  15. In Tables 4, where is the values of reference ascorbic acid.
  16. Table 5 is not mentioned in the results and discussion section.
  17. It is better the tables 4 and 5 will be before fig. 6
  18. Line 380, remove M. pulegium extract was analyzed
  19. Why the order of biological studies in the results and discussion section is differ than that in the methods section.
  20. All references are not unified.